# Investigation of Machine Learning Methods for Prediction of Measured Values of Atmospheric Channel for Hybrid FSO/RF System

**Maroš Lapčák \***, **Ľuboš Ovseník**, **Jakub Oravec** **and Norbert Zdravecký**

Department of Electronics and Multimedia Communications, Technical University of Košice,
04000 Košice, Slovakia; lubos.ovsenik@tuke.sk (Ľ.O.); jakub.oravec@tuke.sk (J.O.);
norbert.zdravecky@tuke.sk (N.Z.)
**\*** Correspondence: maros.lapcak@tuke.sk; Tel.: +421-(55)-602-4277

**Abstract:** This research paper addresses the problems of fiberless optical communication, known as free space optics, in predicting RSSI (Received Signal Strength Indicator) parameters necessary for hard switching in a hybrid FSO/RF (Free Space Optics/Radio Frequency) system. This parameter is used to determine the intensity of the transmitted signal (in our case, a light beam) from one FSO head to another. Since we want to achieve almost 100% reliability, it is important to know the parameters of the transmission environment for the FSO and RF lines. Each of them has its limitations and, as a result, a weather monitoring station is required. The FSO is mostly affected by fog and the concentration of particles in the air, while the RF line is affected by rain and snow. It is precisely due to these influences that it is necessary (based on the mentioned RSSI parameter) to switch using the hard switching method from the primary FSO line to the backup RF line by correctly predicting this value. If the value of the RSSI parameter falls below the critical level—42 dBm—the system automatically switches to the backup RF line. There are several ways we can predict this parameter. One of them is machine learning methods such as decision trees. Our research focused on the prediction of the RSSI parameter, the methods of decision trees and decision trees using the AdaBoost regressor. Since we want to correctly predict the RSSI parameter, it is also necessary to choose the right way to predict it based on the recorded weather conditions. If we want to correctly use the hard switching method in hybrid FSO/RF systems, it is necessary to choose the correct method of predicting the RSSI parameter, which serves as an indicator for switching from the primary FSO line to the secondary RF line. Therefore, we decided to investigate methods of machine learning—the decision tree and the decision tree with the use of an AdaBoost Regressor. The main benefit of this paper is the improvement of existing machine learning methods (decision trees and decision trees using the AdaBoost regressor) for the correct prediction of the RSSI parameter for the needs of hard switching in a hybrid FSO/RF system. The method chosen in this manuscript has very good results. As can be seen in the attached graphs, over a longer period and using correctly selected training data, it is possible to achieve ideal results for the prediction of the RSSI parameter. The tables also show the effectiveness of the prediction, and the fact that it is best to train on either the first- or third-minute data. In the future, it would be appropriate to implement weather prediction or to consider other methods, such as random forests or neural networks.

**Keywords:** FSO; hybrid FSO/RF; machine learning; RF link; weather conditions

## 1. Introduction

Free space optics (FSO) is a technology that uses modulated optical beams to provide wireless data transmission [1]. Communication in this system can take place over short, medium, and also long distances. This means that use is possible for a few meters as well as up to 16 km [2]. A laser is used to create the optical beam, which means that it

is necessary to have a clear line-of-sight between the devices communicating in this way. This laser is housed in a head that can operate in full duplex mode. Thus, one head is both receiving and sending. Free environment, as a term, can be understood, for example, as air or a vacuum. FSO systems work with wavelength bands from 780 to 1600 nm [3]. This means that there is no need to lease licensed frequency bands and it is also not necessary to convert the optical beam into electrical pulses as in the case of radiofrequency (RF) systems [4]. This is a huge advantage for this system. However, the biggest advantages are the large bandwidth and high data rate compared to conventional RF systems. Other advantages include ease of installation and commercial availability. Since FSO systems use an optical beam for communication, the beam angle is very narrow, which prevents intentional eavesdropping, as in this case the connection will be interrupted. Due to these advantages, these systems are used for various applications, such as aeronautical and satellite communications, temporary but also permanent connections between buildings, and so on [2].

It is standard that communication systems must achieve a certain level of reliability. The level of this reliability must be 99.99999% for the use of communication systems. It is affected by three key factors. The first factor is the reliability of the wire and connector, by means of which data is fed to the FSO head. The second factor affecting the availability of the FSO system is the reliability of the electrical and optical components inside the head [5]. These two factors are part of fiber optics, which uses the same wires and connectors as when connecting FSO heads, and at the same time these heads use the same electrical and optical components as those used for active elements in fiber optics. The last and most important factor in the availability of the FSO system is the influence of the transmission environment, as fiber optic connections use fibers made of glass to transmit the optical beam. This means that this factor mostly affects the availability of this system and, thus, its application in the real environment [6].

In this paper, we will focus on machine learning methods (decision trees and decision trees using the AdaBoost Regressor), which will predict the RSSI parameter for the needs of hard switching between FSO and RF lines in a hybrid FSO/RF system. The hard switching method mainly involves lower energy costs associated with the operation of the system, as in this case they do not have to transmit on both lines at the same time. Therefore, it is necessary to choose the right method of machine learning and, above all, to achieve the best possible prediction efficiency. This method was chosen based on the simplicity of implementation into the existing hybrid FSO/RF system at our laboratory and, especially, because of its advantages. The main one is the consideration of all the possible decision outcomes and the tracking of every possible path to the result, and this machine learning method comprehensively analyzes the possible consequences in each branch and identifies decision nodes where further analysis is needed.

### 1.1. Environmental Impact on FSO Communication System

The impact of the environment on the beam propagation may reduce the availability of the FSO system, mainly due to inhomogeneity. This inhomogeneity is mainly due to the different composition over the entire length. As a result, the beam is degraded and, thus, the quality of the received optical signal is reduced. This means that a change in weather can cause a connection deterioration and, in some cases, even a loss of communication [6]. Here is a list of several influences that have a major impact on the quality of the transmitted signal:

- Visibility;
- Temperature;
- Humidity;
- Wind speed;
- Rain;
- Fog;
- Concentration of particles in the air;

- Haze;
- Atmospheric turbulence and scintillation;
- Molecular absorption and scattering [7,8].

*1.2. Hybrid FSO/RF System*

As the name suggests, the hybrid FSO/RF system consists of two systems, as shown in Figure 1. One is the FSO system and the other is the RF system. For hybrid systems such as this, we distinguish between primary and secondary systems. The primary system in this case is the FSO system, as it provides much higher transmission speeds than conventional RF systems. For this reason, the RF system is a secondary system that ensures redundancy of the FSO system in the event of adverse conditions for the primary system. Precisely because of redundancy, hybrid systems use systems with different methods of information transmission. This combination was chosen deliberately, as the biggest problem with the FSO system is fog, and this atmospheric phenomenon does not have a major impact on the operation of RF systems [9]. In this case, it is necessary to consider the transmission speed of individual systems. As the FSO system achieves speeds of a few Gbps and RF only a few hundred Mbps, it is also necessary to consider compensating for these losses in transmission speed. In our hybrid FSO/RF system, we do not monitor speed comparisons, but the availability and reliability itself (99.99999%) by a combination of a primary FSO line and a backup RF line.

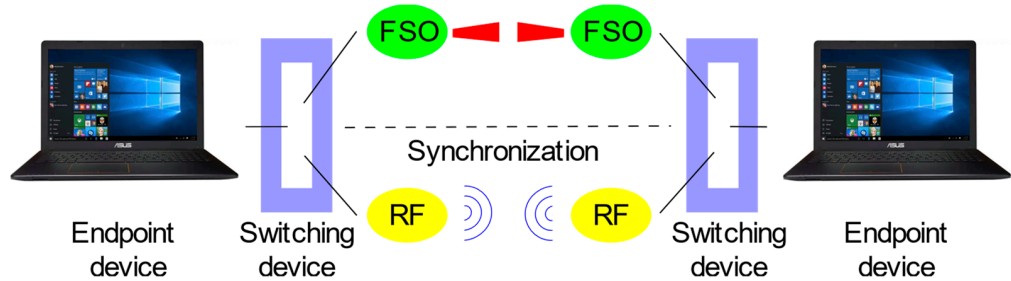

**Figure 1.** Example of hybrid FSO/RF system.

## 2. Machine Learning

Decision trees belong to the widely used machine learning methods that find their application in both classification and regression tasks. In principle, it is a hierarchical, multi-stage binary decision-making system in which the fulfillment/non-fulfillment (if/else) of decision criteria or conditions is gradually evaluated until we reach an accepted class or solution [10]. The decision-making process proceeds from the root of the tree gradually through the individual nodes, which form the branches of the tree with leaves. Although there are several different types of decision trees, the creation of tree structures is mostly governed by the fact that the decision criteria in the individual nodes are arranged according to information importance [11]. The flag or criterion that has the most weight and allows to best separate the input data into two binary classes (yes/no) becomes the root of the tree. The other nodes are gradually made up of the remaining criteria with less weight, with each node creating two binary descendants. The decision-making process follows the individual branches of the tree until we reach the required class or solution (leaf) [12].

The decision tree algorithm method is used to solve regression and classification problems [13]. The result of using a decision tree is an educational model that can be used to predict the value or class of the resulting variable using the way individual learning rules derived from the training data are learned [14]. These rules generally represent training on a specific set of data that the system already knows. It should be noted that a set of training data is needed to train this algorithm. This is made up of several cases $P = \{P_1, P_2, \ldots, P_n\}$, each of which is predefined by a vector with $k$ elements of the tested properties, i.e., the output of the variables $X = \{X_1, X_2, \ldots, X_n\}$, which can be expressed either by number

or word. A decision tree is a tree-structured classifier, where internal nodes represent the properties of a data set, branches represent decision rules, and each leaf node represents a result. There are two nodes in the decision tree, which are the decision node and the leaf node. Decision nodes are used for any decision and have multiple branches, while leaf nodes are the output of these decisions and do not contain any other branches. Decisions or testing is made based on the characteristics of the data set [15].

$$P = \begin{bmatrix} x_{11} & x_{12} & \dots & x_{1k} & y_1 \\ x_{21} & x_{22} & \dots & x_{2k} & y_2 \\ \dots & \dots & \dots & \dots & \dots \\ x_{n1} & x_{n2} & \dots & x_{nk} & y_n \end{bmatrix} \tag{1}$$

When creating a decision tree, the set of input training data is divided into subsets that characterize the input variables in the next steps. In the division process, recursion is used until the termination condition is met. In the analysis of the atmospheric channel for the hybrid FSO/RF system, a system of input cases (variables) $X$ was extensively designed together with the corresponding variables $y$, which are at the output. For hybrid FSO/RF systems, the output variable $y$ is the RSSI value, based on which the transmission lines are switched. The training matrix of the input variables is shown below.

$$X = \begin{bmatrix} TL_1 & TE_1 & KC_1 & V_1 & VL_1 & \dots & RV_1 \\ TL_2 & TE_2 & KC_2 & V_2 & VL_2 & \dots & RV_2 \\ \dots & \dots & \dots & \dots & \dots & \dots & \dots \\ TL_n & TE_n & KC_n & V_n & VL_n & \dots & RV_n \end{bmatrix} \tag{2}$$

The individual columns represent the measured values using the weather monitoring station. $TL$ is barometric pressure [hPa], $TE$ represents air temperature [°C], $KC$ represents particle concentration [mg/m³], $V$ is visibility [m], $VL$ represents ambient humidity [%], and $RV$ is wind speed [m/s]. The matrix of the output variable $y$ (target) represents the received optical power.

$$y = \begin{bmatrix} y_{RSSI.1} & y_{RSSI.2} & \cdots & y_{RSSI.n} \end{bmatrix}^T \tag{3}$$

Figure 2 shows an example of generating a decision tree algorithm. If we apply this to solve the problem of switching a hybrid FSO/RF system, then it consists of a set of cases to determine whether a person can work or not. In this case, root node X1 represents the fatigue state and decision node X2 represents the time of day. Thus, for the FSO/RF system, it is: X1—particle concentration (KC); X2—visibility (V).

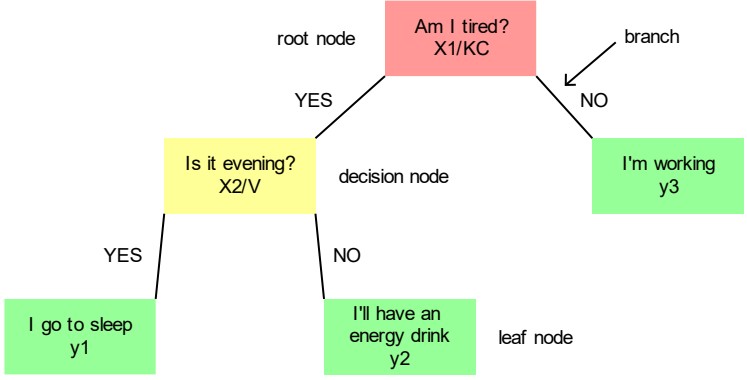

**Figure 2.** Example of a decision tree.

To calculate $y_d$, the equation would look like this:

$$y_d = \begin{bmatrix} y_{igotosleep} & y_{I'llhaveanenergydrink} & y_{I'mworking} \end{bmatrix}^T \tag{4}$$

*2.1. Decision Trees Using AdaBoost Regressor*

The AdaBoost algorithm iteratively modifies the learning set based on the classifiers already created. Those examples that were misclassified by the last model created will be given more weight in the next iteration. This means that when learning a new classifier, more emphasis will be placed on these examples, and so this new classifier will focus on the part of the concept in which the previous classifier failed. A weighted choice across all classifiers in sequence is then used to classify the new examples [16,17].

*2.2. Received Signal Strength Indicator*

The strength of the received optical power RSSI is an important decision indicator in the process of hard switching of a hybrid FSO/RF system [18]. Modern FSO/RF systems switch to RF transmission mode after exceeding the critical low received optical power limit on the receiving side. This information must be distributed and synchronized on both sides of the line. The hybrid system switches back to optical mode if the value of the current RSSI is above the critical level for a certain time [19]. The operating frequencies of commercially deployed RF lines are usually around 60 GHz (unlicensed band), which means that the transmission speed of such a system is disproportionately lower than that of an optical line [20].

However, there is an effort to make a reliable prediction of the development of the RSSI parameter, based on which it would be possible to identify the necessary communication time of the hybrid system more accurately in the RF mode. Effective reduction in the operating time of the hybrid FSO/RF system will increase the maximum use time of the high transmission speed of the FSO line, while maintaining the condition of continuous communication between the two nodes of the hybrid FSO/RF system [21]. The optical beam emitted from the FSO head is naturally attenuated by passing through the atmospheric channel, depending on the current nature of the weather conditions. The received RSSI optical power detected on the optical receiver can in fact be interpreted as a function of a set of parameters describing the nature of the weather conditions. In the present system, the parameters of temperature, humidity, and concentration of particles in the air, visibility, barometric pressure, and wind speed are measured and recorded in real time [22].

## 3. Recording of Atmospheric Conditions Using a Weather Monitoring Station to Predict RSSI Value

Communication standards place a high demand on FSO systems throughout the year. The availability and reliability of FSO systems is an overall problem and in solving it, is necessary to analyze enough auxiliary input physical quantities, which characterize the atmospheric conditions along the entire transmission path in real time.

Atmospheric data collection is provided by a weather station and sensors working on a Raspberry Pi 3 microcomputer. The monitoring station together with the hybrid FSO/RF system is shown in Figure 3.

Monitoring the weather conditions between the two heads of FSO systems is an important task in determining and evaluating the reliability and availability of FSO systems. An important condition for securing an optical connection using an FSO line is direct visibility between the receiver and the transmitter. The full-duplex principle applies to this type of communication; hence, each device is a transmitter and a receiver at the same time. The list of monitoring station sensors is written in Table 1.

The weather station created in this way is located directly at the transmitter/receiver; thus, it does not record weather conditions directly on the entire route. On the other hand, the subsequent monitoring system can record the entire transmission path, but not for the entire duration of the communication.

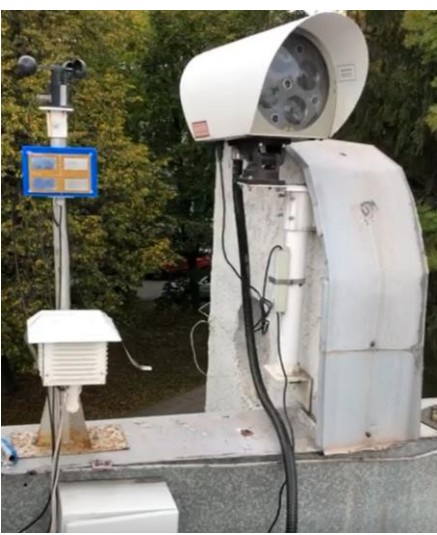

**Figure 3.** FSO head FlightStrata 155E with backup RF line and weather monitoring station.

**Table 1.** List of sensors.

| Physical Quantity | Indication | Sensor Name | Units |
|---|---|---|---|
| Signal Strength | RSSI | Flight Strata 155E | dBm |
| Visibility | VI | miniOFS | m |
| | TE1 | gp2y1010au | °C |
| Temperature | TE2 | DS18B20 | °C |
| | TE3 | DHT22 | °C |
| Concentration of particles in the air | KC | gp2y1010au | $\mu g/m^3$ |
| Wind Speed | RV1 | Anemometer | m/s |
| Wind Voltage | RV2 | Anemometer | m/s |
| Humidity | VL | DHT22 | % |
| Pressure | TL | BMP183 | hPa |

### 3.1. Recording of Atmospheric Conditions Using Unmanned Aerial Vehicle (UAV)

This sensor system plays a significant role in the deployment of FSO technology in critical situations. If necessary, this technology can be used within minutes to analyze and map the appropriate location of the FSO head. It will allow us to quickly detect an unfavorable environment for this technology and the subsequent deployment of a secondary RF communication line. It is a technology suitable for rapid deployment in the field. The sensor system can also be used in hard-to-reach places or life-threatening places without the need to shut down these dangerous factors.

The sensor system focuses on three basic factors affecting the ability to use FSO. It will detect humidity and ambient temperature, as at high temperatures near water areas there is a high probability of mist and water particles in the air absorbing or scattering the laser beam, resulting in inefficient or even impossible use of FSO technology in the area. Furthermore, the system detects the presence of dust particles in the air. These particles have the most significant effect on the FSO capability of the head. If there is a high occurrence of polluting particles in the air, the beam can be greatly attenuated or even completely interrupted, which is the most unfavorable scenario. The most important part of this system is the ability to map the environment and then plot it in Google Earth with the appropriate air quality values. The actual connection of the sensor system can be seen in Figure 4.

From information about air pollution obtained by drone survey flights in several areas, we obtained sufficient data to analyze air quality and its impact on the FSO communication line. The acquisition of the necessary data took place in two localities. The first location was the premises of the Technical University in Košice; several reconnaissance flights took place

here. The survey was conducted to design the location of FSO technology as an extension of the existing communication network. The first and second columns describe latitude and longitude, and the third to fifth columns describe the measured value of temperature, humidity, and dust particles in the air. Temperature was measured in degrees Celsius (°C), humidity in percent (%), and air pollution in micrograms per cubic meter ($\mu g/m^3$). The Google Earth Pro location data processing program was used to display the research results on the map. In Figure 5 it is possible to see the whole assembly of the UAV device together with the weather monitoring station and Figure 6 shows the legend to the map shown in Figure 7.

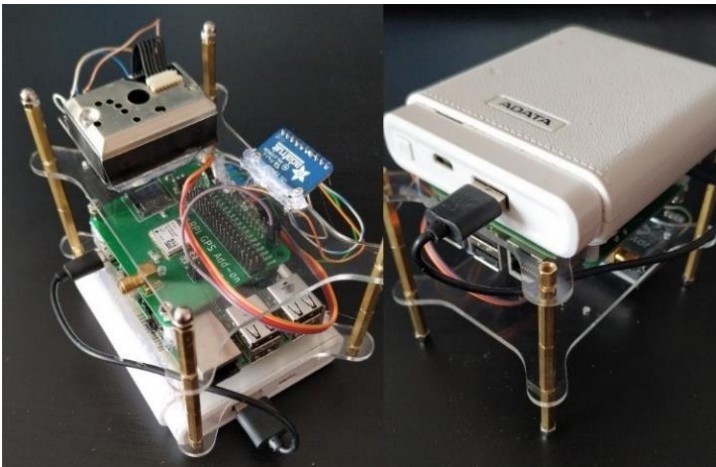

**Figure 4.** Connections of the sensor system located on the UAV.

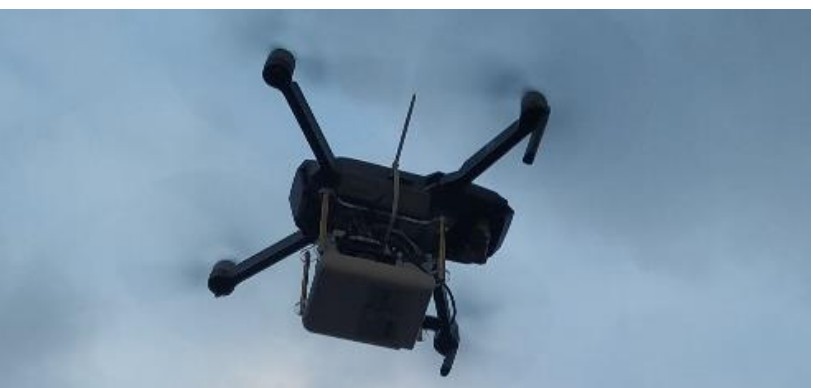

**Figure 5.** UAV device with sensor system.

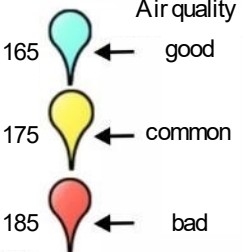

**Figure 6.** Legend of air quality symbols.

　　　The output values were analyzed to identify a suitable place for storing the FSO head and the subsequent creation of a high-speed line on the premises of the Technical University in Košice (TUKE). From these data, there is no above-average air pollution in this location. The temperature during the measurement averaged 21.3 °C and 40% humidity.

Measurements of air pollution and the presence of airborne particles were successful. No disturbing air quality data were found on the premises of the TUKE. The values of particles in the air did not exceed 800 μg/m³, which means only average pollution. The number of measured values in space is easily visible on the map in Figure 7. The individual points shown on the map represent the measurement points. The measurement was performed every 5 s and then assigned to the position based on continuity of time. Table 2 shows sample data for processing in graphical Google Earth.

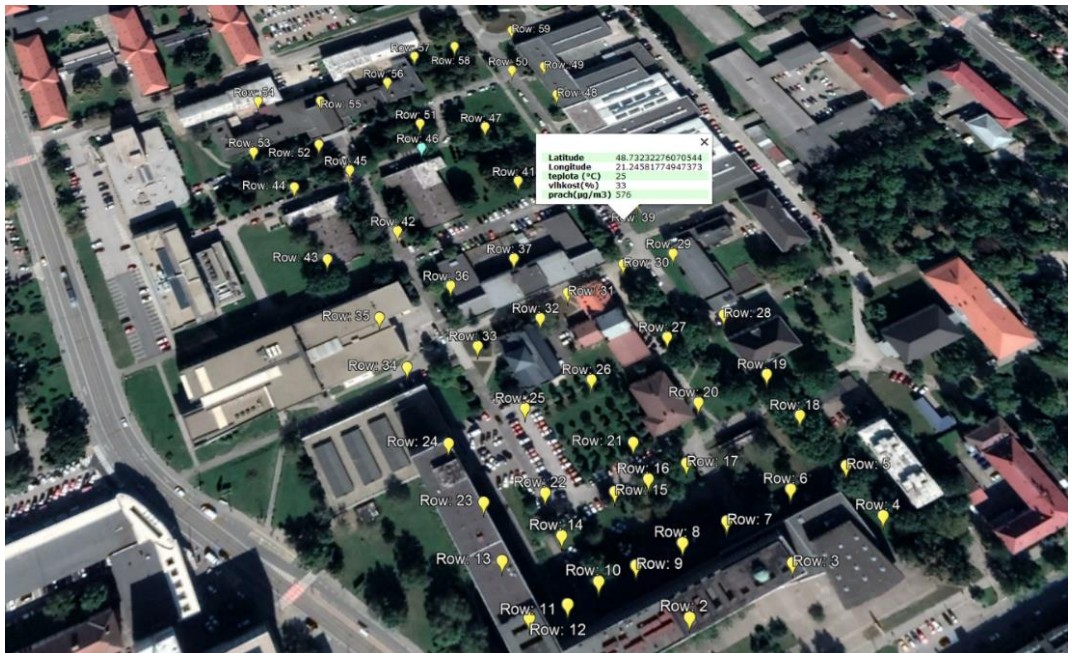

**Figure 7.** Map of the mapped area in the TUKE locality with recorded values.

**Table 2.** Sample data suitable for analysis in a graphical interface.

| Latitude | Longitude | Temperature (°C) | Humidity (%) | Particles in the Air (μg/m³) | Icon |
|---|---|---|---|---|---|
| 49.1476795377 | 19.2358069399 | 10 | 30 | 129 | 165 |
| 49.1480705071 | 19.2363364994 | 7 | 37 | 482 | 175 |
| 49.1485925283 | 19.2357429895 | 6 | 35 | 551 | 175 |
| 49.1478324859 | 19.2351676248 | 5 | 38 | 506 | 175 |
| 49.1478180217 | 19.2342893291 | 4 | 39 | 535 | 175 |
| 49.1487775524 | 19.2348703706 | 4 | 43 | 432 | 175 |
| 49.1473503714 | 19.2348536109 | 3 | 44 | 432 | 175 |
| 49.1489112933 | 19.2352287229 | 2 | 48 | 375 | 175 |
| 49.1469179250 | 19.2359150177 | 2 | 49 | 424 | 175 |
| 49.1464293509 | 19.2348498906 | 2 | 52 | 420 | 175 |

The main disadvantages or limitations of our proposed system are the congestion of the system and its computational complexity. If we use more data, the prediction calculation will take much longer than if we use less data. Therefore, it is desirable that only the data necessary for RSSI parameter prediction be selected to select the input amount of data. As a possible solution, it would be possible to use only data from a given period (for example, a day) and not to use data recorded by a weather station from a period of one week, two weeks, a month, etc.

*3.2. Measurrement of the RSSI Parameter*

The values of the received RSSI optical power on both sides of the FSO line can be visualized using software from the manufacturer Flight Manager PC in the form of a graph. The Flight Manager PC software records RSSI in specially organized files with the extension *.dat (application/octet-stream; charset = binary) from which it is possible to visualize the recorded RSSI later.

In the case of the FSO of the FlightStrata 155E line and its control software, the received RSSI optical power is only informative (a form of visualization). However, this parameter plays an important role in the analysis of availability and reliability. By applying reverse software engineering, the *.data files (output of the Flight Manager PC software) were subjected to a detailed analysis with the aim of continuously recording the value of the received optical power with the corresponding time record.

The recorded data was updated and overwritten in a file every time it was turned on, and after the measurement was finished, it was sent to the Raspberry Pi 3 microcomputer, where the weather conditions were also recorded. Subsequently, they were entered into the common MySQL database.

Since the values of the RSSI parameter were recorded using software compatible with our FlightStrata 155E head, we can say that the RSSI parameter has the same parameters (in terms of wavelength, etc.) as the FSO head after correct setting.

*3.3. Square Root Mean Square Error*

If we want to calculate the measurement accuracy, we need to determine the standard deviation. This is often used to determine the differences between values (real and predicted) in machine learning methods to predict values. We chose square root mean square error (*SqRMSE*) [23,24].

$$SqRMSE = \sqrt{\frac{1}{n}\left(RSSI_r - RSSI_p\right)^2}\ [\text{dBm}] \tag{5}$$

Formula (5) represents the square root of the square averages of these differences. $n$ in this formula represents the number of all samples, $RSSI_r$ represents the real values of the $RSSI$ and $RSSI_p$ stands for predicted values. In our case, we decided to calculate this deviation for individual measurements and then averaged it according to the number of total samples. *SqRMSE* is used to compare prediction errors for a particular data set and mainly depends on the scope of the data. The *SqRMSE* value can never be negative. In general, the lower the *SqRMSE* value, the better it is compared to the higher value [23].

**4. Results**

Input data containing information about measurement dates $D$ (21.9.2020 [DMY]), measurement times $C$ (7:52 [HOUR: MIN]), information about received $RSSI$ signal strength ($-32.375$ [dBm]), current temperature $TE1$ (17.3 [°C]), pressure $TL$ (990.52 [hPa]), current temperature $TE2$ (15.4 [°C]), air humidity $VL$ (49.099998 [%]), current temperature $TE3$ (11.187 [°C]), wind speed $RV$ (0.403695 [m/s]), wind speed $SV$ (0.396887 [m/s]), particle concentration $KC$ (13 [mg/m$^3$]), and visibility $VL$ (4000 [m]) should be stored in a CSV file, where the individual values in the line are separated by a decimal point. The decimal point, therefore, represents the column separator. The individual rows are arranged one below the other, forming an input data set. The measurements were performed in a certain period with a minute interval between individual measurements. This set of input data was obtained from the weather station of the Technical University in Košice located on the PK13 building and contains almost 120,000 records from the dates from 21.9.2020 to 13.12.2020. An example of measured data is shown in Figure 8.

```
21.9.2020,7:52,-32.375,17.3,990.52,15.4,49.099998,11.187,0.403695,0.396887,13,4000
21.9.2020,7:53,-25,17.35,990.505,15.55,49.1999985,11.85,0.415297,0.397262,13,4000
21.9.2020,7:54,-25,17.55,990.525,15.4,48.950001,11.281,0.415297,0.397262,11.5,4000
21.9.2020,7:55,-25,17.55,990.56,15.55,48.8999995,11.35,0.415297,0.397262,12,4000
21.9.2020,7:56,-25,17.4,990.545,15.7,48.25,11.375,0.4133635,0.3971995,13,4000
21.9.2020,7:57,-25,17.4,990.54,15.55,48.049999,11.375,0.415297,0.397262,13,4000
21.9.2020,7:58,-25,17.5,990.605,15.55,48.3500005,11.375,0.4133635,0.3971995,13,4000
21.9.2020,7:59,-25,17.5,990.55,15.4,48.400002,11.312,0.778824,0.4090125,14,4000
```

**Figure 8.** Example of formatting input data in a CSV file.

We can also imagine the input data in the form of the matrix shown below:

$$
M = \begin{bmatrix}
D_1 & C_1 & RSSI_1 & TE1_1 & TL_1 & TE2_1 & VL_1 & TE3_1 & RV1_1 & RV2_1 & KC_1 & VI_1 \\
D_2 & C_2 & RSSI_2 & TE1_2 & TL_2 & TE2_2 & VL_2 & TE3_2 & RV1_2 & RV2_2 & KC_2 & VI_2 \\
\ldots & \ldots & \ldots & \ldots & \ldots & \ldots & \ldots & \ldots & \ldots & \ldots & \ldots & \ldots \\
D_n & C_n & RSSI_n & TE1_n & TL_n & TE2_n & VL_n & TE3_n & RV1_n & RV2_n & KC_n & VI_n
\end{bmatrix}
\tag{6}
$$

From the initial look at the set of input data, it was clear that not all parameters affect the RSSI value to the same extent. Therefore, we decided to use correlation to determine the magnitude of the impact of individual parameters on the RSSI value.

To calculate the correlation, a program in the Python programming language was created. Three correlation methods were used for the calculation—the method using Pearson's correlation coefficient ($\rho_p$), then the method with Spearman's correlation coefficient ($\rho_s$), and finally the method with Kendall's correlation coefficient ($\rho_k$). We always compared the column of parameter values of the received RSSI signal with the column of one of the parameters of climatic conditions. All functions return two values—the first being the correlation coefficients, which were stored, and then the values used to test the hypothesis, which are not used in our proposal, so they are discarded. The outputs are rounded to five decimal places and calculated for all correlation functions.

The results of Pearson's method, which compares the linear dependence between two sets of data, in our case specifically between the RSSI parameter and individual parameters of climatic conditions, appeared to be the best. The correlation coefficient is between $-1$ and $+1$ where 0 represents no correlation and $-1$ and $+1$ represent the same linear relationship. A positive number means that with a gradual increase in the RSSI value, a specific parameter of climatic conditions increases. On the contrary, a negative number means that as the RSSI value increases, the specific parameter of climatic conditions decreases.

From Table 3, at the Pearson correlation coefficient, we see that the intensity of the received RSSI signal is most affected by visibility, particle concentration, and temperature, which are closely related to rain, snow, or fog. For this reason, we tried to train individual prediction models using only these specific parameters or by averaging individual temperatures, wind speeds, and strengths. We were mainly interested in reducing the required computing power, as with the gradual growth of the database of input values, small details will fundamentally affect the required performance. That is why we also started experimenting with using data not from every minute but every second minute, third, fifth, or tenth minute. We could afford it, as the weather usually does not change from minute to minute, but is always a longer process. In our database with 120,000 records, we saw the use of every second or third row as the most suitable, when the deviations in the prediction of individual algorithms were still in a low range.

As the first machine learning model, we decided to try decision trees. Decision trees are based on the principle of a hierarchical multi-stage binary decision-making system, in which the fulfillment or non-fulfillment of individual decision-making conditions is gradually evaluated until we reach an acceptable solution. Furthermore, it was necessary to divide the input data set into algorithm training data and test data. We chose a ratio of 80% training data and 20% testing. We also conducted experiments with a ratio of 70:30, but the results of the prediction success were slightly weaker. To optimize the complexity of the

training algorithm and the amount of data generated during training, we decided to first try to calculate the average from the data obtained from three temperature sensors, as well as two wind speed sensors, and then support only these averaged values in the training algorithm. In the next step, we performed an arithmetic mean for the temperature data, where we added the values from all three sensors and divided them by three, and for the wind speed data, where we added the data from both sensors and divided them by two.

**Table 3.** Results of correlation of RSSI parameter with individual parameters of climatic conditions.

| Parameter Y | $\rho_p$ (RSSI, Y) | $\rho_s$ (RSSI, Y) | $\rho_k$ (RSSI, Y) |
|---|---|---|---|
| RSSI | 1.00000 | 1.00000 | 1.00000 |
| Temperature 1 | 0.16330 | −0.18181 | −0.13946 |
| Pressure | 0.06568 | 0.16756 | 0.12736 |
| Temperature 2 | 0.15377 | −0.18674 | −0.14321 |
| Humidity | −0.02592 | −0.19099 | −0.15497 |
| Temperature 3 | 0.17732 | −0.18033 | −0.13849 |
| Wind speed | −0.01588 | 0.14626 | 0.11344 |
| Wind voltage | −0.01594 | 0.14622 | 0.11343 |
| Particles in the air | 0.17781 | −0.01625 | −0.01391 |
| Visibility | 0.24912 | −0.12074 | −0.10815 |

It is known that decision trees improved using the AdaBoost algorithm are characterized by higher accuracy than pure decision trees. Using AdaBoost regression, we managed to achieve a success rate of up to 92%, which is 11% more than with the common decision tree algorithm (sample of 120,000 records, data from every minute). However, the complexity of the computational algorithm is many times higher with AdaBoost regression than with decision trees. With 120,000 records, the training process took up to 1 min 17 s compared to 0.667425 s for decision trees.

We have constructed the following tables and graphs based on several outputs from various data configurations and programs. Tables 4–6 compare the success and length of the decision tree and AdaBoost algorithm training process for data from every 1st, 2nd, 3rd, 5th, and 10th minute. We see that even when using data from every second or third minute, the prediction efficiency still reaches high values. The difference between Tables 4 and 5 lies in the different value of the n_estimators parameter of the AdaBoostRegressor function. Table 4 shows the values for the value n_estimators = 300, while Table 5 shows the values for the value n_estimators = 150. As we can see from the values, the effectiveness of the AdaBoost prediction is practically unchanged, while the time required for training has doubled in any case. As a result, we have determined that limiting this value to 150 will not adversely affect the results. At values lower than 150, there was already a gradual decrease in the effectiveness of the prediction, for example, with the value of n_estimators = 130 and data from each minute, we achieved a success rate of only 89% compared to the original 92%.

**Table 4.** Comparison of individual algorithms (120,000 records, n_estimators = 300, training ratio 80:20)—data used: temperature 1, 2, and 3; pressure; humidity; wind speeds 1 and 2; particle concentration, visibility.

| Training with Data from Each | Length of the Decision Tree Training Process | Training Process Length (AdaBoost) | Decision Tree Prediction Efficiency | AdaBoost Prediction Efficiency |
|---|---|---|---|---|
| 1 min | 0.68 s | 2:27 min | 81% | 92% |
| 2 min | 0.34 s | 1:06 min | 76% | 86% |
| 3 min | 0.23 s | 0:41 min | 78% | 88% |
| 5 min | 0.14 s | 0:22 min | 56% | 73% |
| 10 min | 0.08 s | 0:10 min | 15% | 56% |

**Table 5.** Comparison of individual algorithms (120,000 records, n_estimators = 150, training ratio 80:20)—data used: temperature 1, 2, and 3; pressure; humidity; wind speeds 1 and 2; particle concentration, visibility.

| Training with Data from Each | Length of the Decision Tree Training Process | Training Process Length (AdaBoost) | Decision Tree Prediction Efficiency | AdaBoost Prediction Efficiency |
|---|---|---|---|---|
| 1 min | 0.66 s | 1:17 min | 81% | 92% |
| 2 min | 0.32 s | 0:33 min | 76% | 86% |
| 3 min | 0.21 s | 0:21 min | 78% | 88% |
| 5 min | 0.12 s | 0:11 min | 56% | 74% |
| 10 min | 0.06 s | 0:05 min | 13% | 54% |

**Table 6.** Comparison of individual algorithms (120,000 records, n_estimators = 150, training ratio 80:20)—data used: temperature average; pressure; humidity; wind speed average; particle concentration, visibility.

| Training with Data from Each | Length of the Decision Tree Training Process | Training Process Length (AdaBoost) | Decision Tree Prediction Efficiency | AdaBoost Prediction Efficiency |
|---|---|---|---|---|
| 1 min | 0.499773 s | 0:53 min | 78% | 86% |
| 2 min | 0.228143 s | 0:24 min | 69% | 84% |
| 3 min | 0.147259 s | 0:15 min | 71% | 85% |
| 5 min | 0.090056 s | 0:08 min | 58% | 73% |
| 10 min | 0.044001 s | 0:03 min | 0.09% | 62% |

The difference between Tables 5 and 6 is the fact that Table 5 shows the average values of temperatures and wind speeds from several sensors. Although it might seem that the prediction efficiency should not change, as we only averaged similar values from several sensors, from Table 6 we see that this is not the case. It could be caused by the different location of the sensors, their measurement deviation, or the specific properties of the sensors. However, averaging further reduces the time required to train algorithms. The actual values of the RSSI parameter are shown in Figure 9.

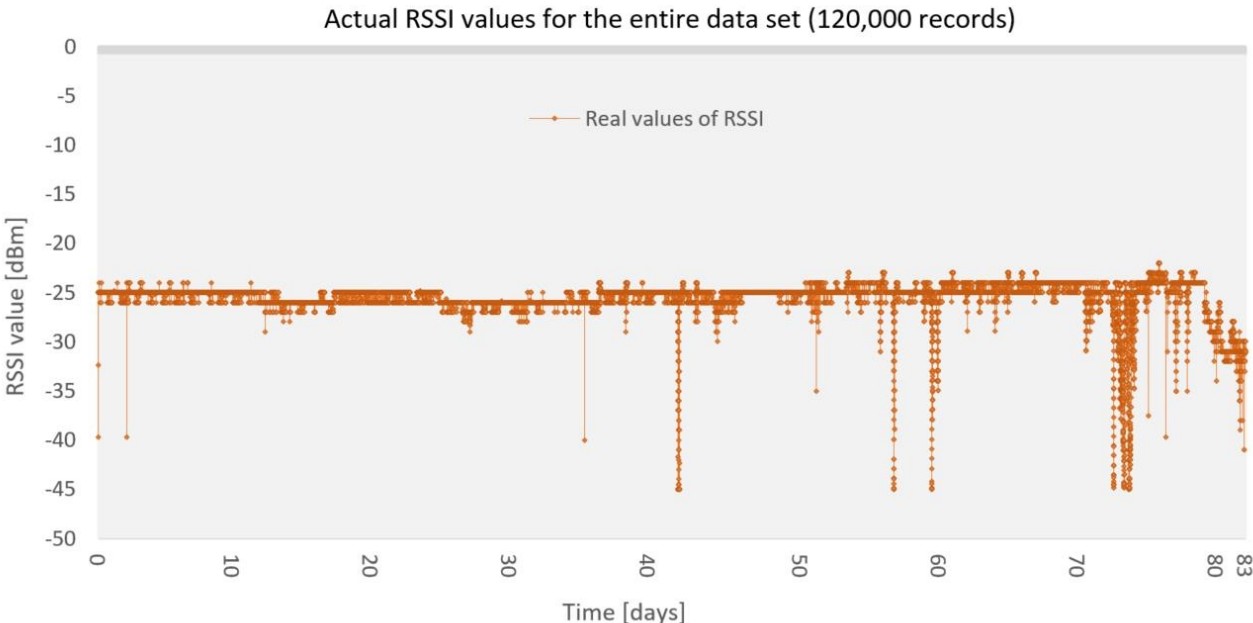

**Figure 9.** Actual values of the RSSI parameter for the entire data set (120,000 records).

At the beginning, we started testing the decision tree method. If we collected a small amount of data (17 November 2020 16:40 to 19:40) and all the measured values, the results were not optimal. The prediction was not of high quality and the value of SqRMSE itself was at the level of 0.393997 dBm. Therefore, we also tried the decision tree method on the same file using the AdaBoost Regressor. In this case, the results were slightly better. The SqRMSE value was 0.384952 dBm. As can be seen in Figure 10, the prediction using decision tree and decision tree using AdaBoost Regressor was practically the same.

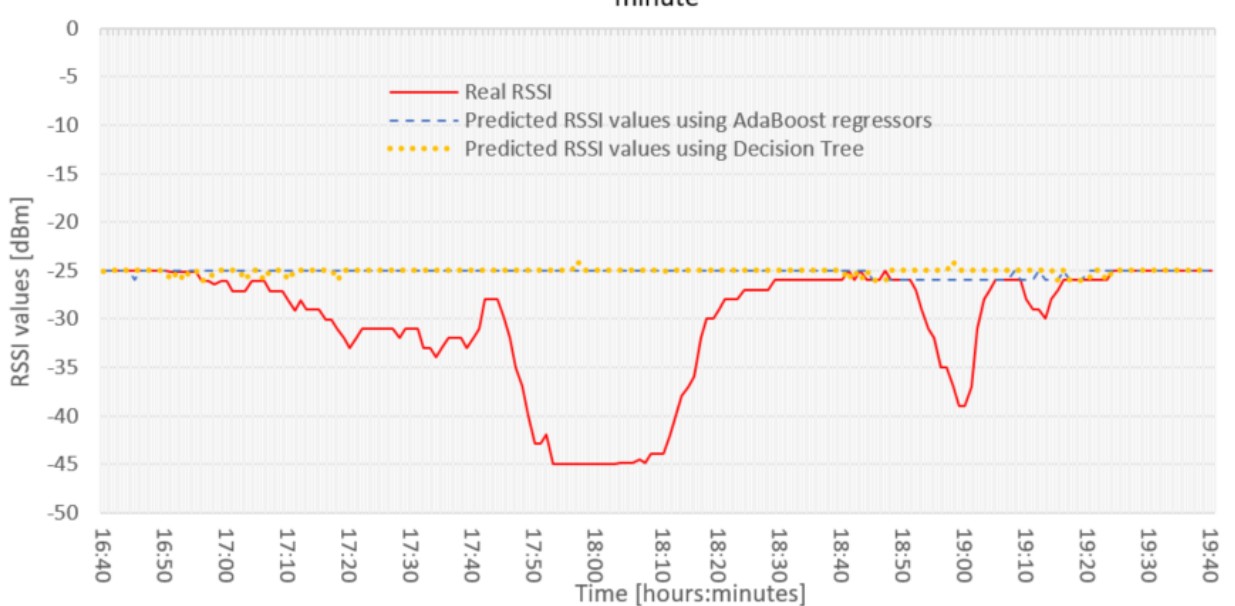

**Figure 10.** Comparison of real RSSI values with values predicted by decision tree (yellow) and values predicted by AdaBoost regressor (blue) trained from data from every 1 min.

In the next case, we wanted to test the accuracy of the prediction from a given data set, from which we selected values from every second and every third minute. These results did not turn out well at all, as the SqRMSE values for every second minute were 0.739108 dBm and for every third minute were 0.969592 dBm. Thus, we can say, for sure, that with a small amount of data, the prediction from every second and third minute is ineffective, as can be seen in Figure 11.

Subsequently, our algorithm was subjected to the testing of several variants, namely averaging temperatures from three sensors, averaging wind speed, selecting only some environmental parameters from the database, etc.

As we can see in the graph in Figure 12, using a smaller set but more variable data, the decision tree model hit at least some values (blue color). AdaBoost improved this model even more, as we can see in Figure 12 (orange color). In this case, however, the SqRMSE value for decision tree was up to 0.503695 dBm and for AdaBoost Regressor 0.414687 dBm.

In the graph of Figure 13, we can see attempts to train the AdaBoost regressor with averaged values of temperature and wind speed. The prediction was not sufficient in this case either, so we can say with certainty that it is not very good to average the values of several physical quantities from several sensors. In this case, the SqRMSE value was 0.406414 dBm, which is, however, comparable to the value using the AdaBoost Regressor from a smaller amount of data than in Figure 10.

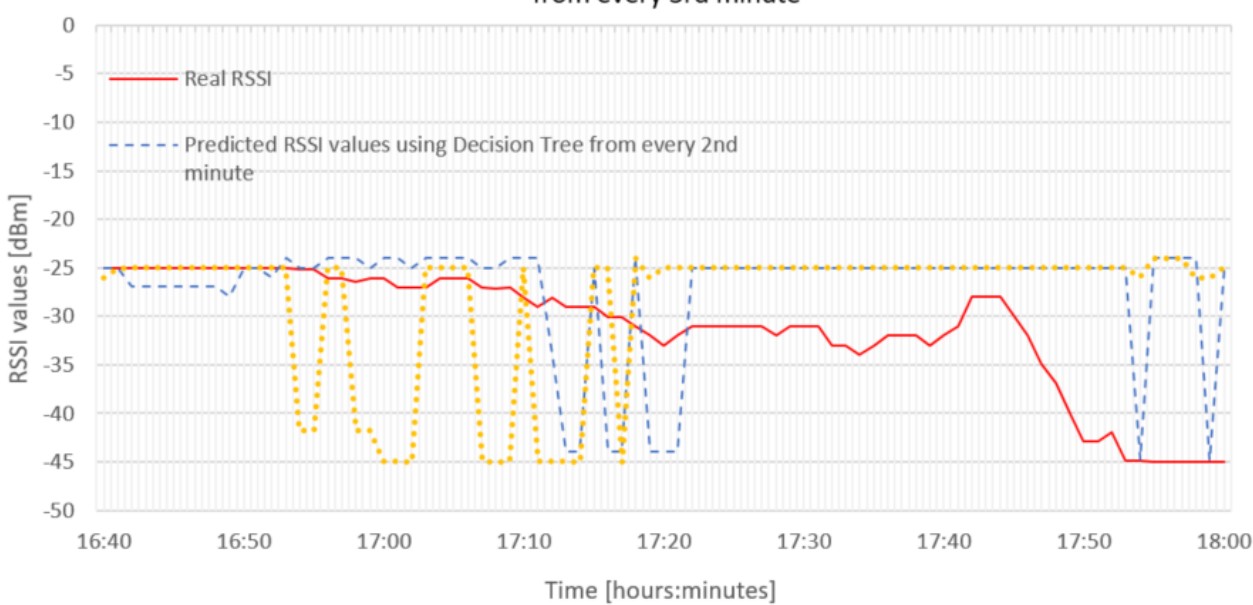

**Figure 11.** Comparison of real RSSI values with values predicted by decision tree trained from data from every 2 min and values predicted by decision tree regressor trained from data from every 3 min.

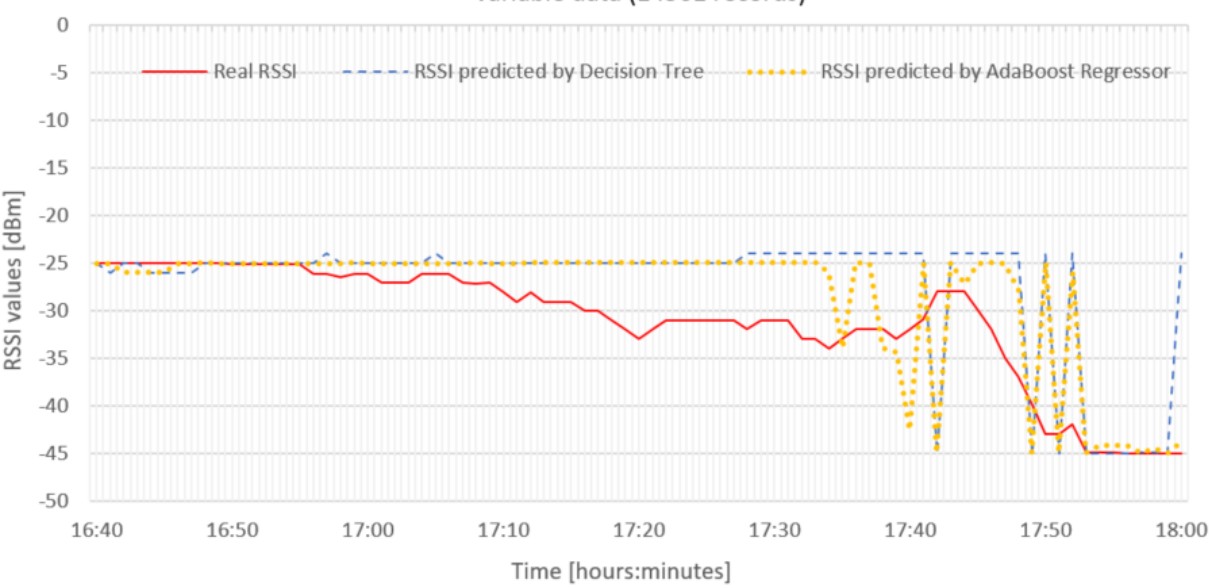

**Figure 12.** Comparison of real RSSI values with values predicted by decision tree and values predicted by AdaBoost regressor trained from a large amount of variable data.

We even tried to train the model using the AdaBoost Regressor without data, in which the correlation showed the least dependence (humidity and wind force, later also pressure), but as we can from the graph in Figure 14, this prediction did not turn out the best. In this case, the SqRMSE value was 0.591647 dBm.

We also experimentally tried to extend the reduced training set by a certain amount of data from the month of December. Specifically, we added 4896 records from December to the reduced set (the prediction will be 100% successful), which together created a training set of 10,932 records. We wanted to test whether a certain amount of correct data can

help improve the prediction of surrounding values. The predicted data for the month of December contain 17,445 records, of which 4896 records were correct, the rest was predicted. How the whole thing was conducted can be seen in the pictures from Figures 15 and 16. In these cases, the SqRMSE value was 0.00844 dBm and 0.0140798 dBm. Thus, we can say with certainty that, compared to previous cases where only a small data set was used for prediction, our machine learning model for RSSI parameter prediction is very reliable.

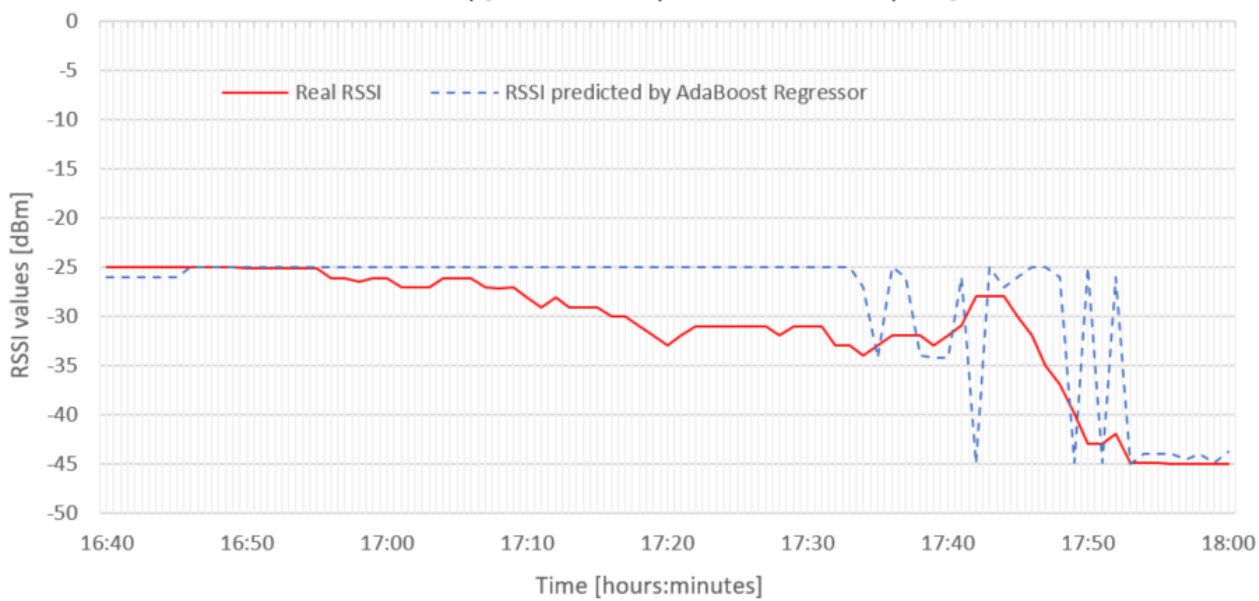

**Figure 13.** Comparison of real RSSI values and values predicted by AdaBoost regressor trained from a data set with averaged values of temperature and wind speed from different sensors.

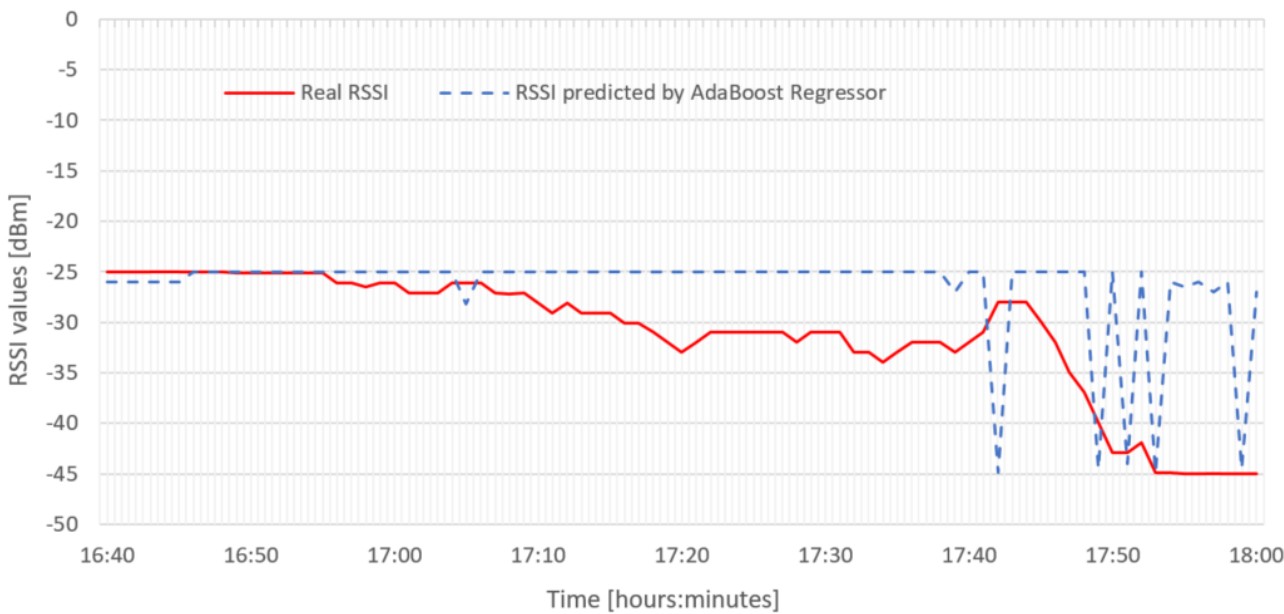

**Figure 14.** Comparison of real RSSI values and values predicted by AdaBoost regressor trained from a data set only with average temperature, pressure, particle concentration, visibility.

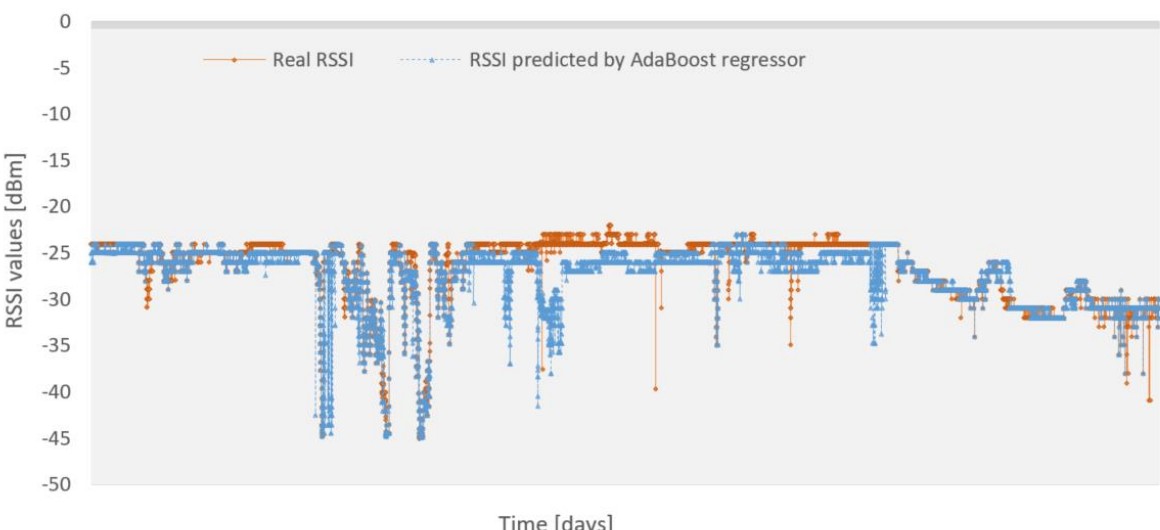

**Figure 15.** Comparison of real RSSI values and values predicted by AdaBoost regressor trained from a larger amount of variable data + a partial amount of real data from the predicted period.

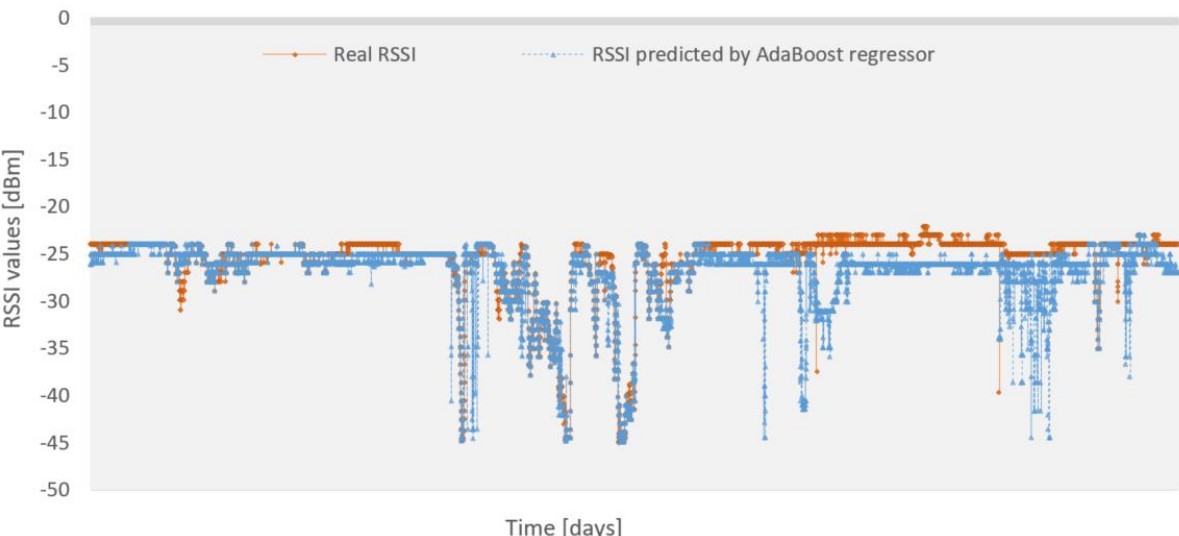

**Figure 16.** Comparison of real RSSI values and values predicted by AdaBoost regressor trained from larger amount of variable data + partial amount of real data from predicted period (4896 values and 10932 records), averaged values of temperature and wind speed.

## 5. Conclusions

The RSSI parameter is one of the most important parameters for hybrid FSO/RF systems. Thanks to it, we can correctly configure the hard switching method and decide which line will transmit at a given time. In this work, two machine learning methods for RSSI parameter prediction were tested. It was a decision tree method and a decision tree method using the AdaBoost Regressor function.

If we want to correctly predict the value of the RSSI parameter, we must correctly understand the dependence of this parameter on atmospheric conditions. These were recorded

using a weather monitoring station located at one of the FSO heads and a UAV device. Based on these parameters, correlations were found between the individual parameters and the RSSI value. Subsequently, we proceeded to implement the machine learning program itself using the method of decision trees. When we used training:test ratio 80:20 data distribution, we achieved high training speed and good results, but in other more detailed tests it was not so successful. The second implemented program was based on the decision tree algorithm improved by AdaBoost regression. The results obtained by this machine learning algorithm showed a higher prediction efficiency than the decision trees themselves, but also the length of training and improvement of such a model was many times higher. We tried to train the models with different input data configurations—we only tried to use data from every second, third, fifth, or tenth minute—and then we analyzed the success of the prediction. In targeted testing of the prediction in highly variable data, we found that the success of the prediction did not reach the desired value, as the set of input training data mostly contained similar values of individual parameters (nice weather) and peaks (rain, snow, fog) occurred in it only to a small extent. Therefore, we tried to train algorithms with a data set containing reduced redundant data. In this case, the success of the algorithms managed to reach a higher level, but it still did not reach optimal values. If we pushed part of the real values from the predicted period into the algorithm, the efficiency of the prediction around these values increased significantly. From this, we concluded that it is very important when training the algorithms to make sure that the input data set is sufficiently variable and covers all possible states to approximately the same extent. This information could serve as a basis for further future research on this issue. To achieve even better results and high efficiency of prediction, it would be appropriate to consider the prediction of weather parameters and, thus, examine the behavior of the decision tree algorithm. Since the decision tree is the basis of the random forest algorithm, in further research we would like to focus on this algorithm and examine its prediction efficiency.

**Author Contributions:** Conceptualization, M.L. and Ľ.O.; methodology, M.L.; software, M.L.; validation, M.L., Ľ.O., J.O. and N.Z.; formal analysis, M.L.; investigation, M.L.; resources, M.L., Ľ.O., J.O. and N.Z.; data curation, M.L.; writing—original draft preparation, M.L.; writing—review and editing, M.L., Ľ.O., J.O. and N.Z.; visualization, M.L.; supervision, Ľ.O. and J.O.; project administration, Ľ.O. and J.O.; funding acquisition, Ľ.O. All authors have read and agreed to the published version of the manuscript.

**Funding:** This work was supported by the research project VEGA 1/0584/20 "Person Monitoring by UWB Sensor Systems Operating in Real Conditions".

**Institutional Review Board Statement:** Not applicable.

**Informed Consent Statement:** Not applicable.

**Data Availability Statement:** The data presented in this study are available on request from the corresponding author.

**Conflicts of Interest:** The authors declare no conflict of interest.

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
