# Peer review of "Investigation of Machine Learning Methods for Prediction of Measured Values of Atmospheric Channel for Hybrid FSO/RF System"

_photonics, doi:10.3390/photonics9080524_

Round 1

Reviewer 1 Report

I think the present form could be considered for publication.

There is a reference error on line 414.

Reviewer 2 Report

The authors propose the strategy of Adabost and Adabostregressors to develop a machine learning strategy to estimate the RSSI signal. The results that they obtain are quite low in terms of performance for all cases. Even in the case that they say that the performance of their system in Figure 12 the Adabost regressos is 0.041, the figure 12 shows that there is a big difference between the real RSSI and the predicted RSSI by the Adabost regressor.

In Figure 15 and 16 where it is assumed that the results are much better is not clear to me such improvement. In any case, I am not convinced that the methodology developed for designing this system be the most appropiate. The signal model of the signal that you want to estimate is not included nor the frames that you use to estimate the RSSI, nor the bandwidth of the signal nor the receiver estructure nor the detection part. 

Reviewer 3 Report

I have checked the whole manuscript. The author has corrected the manuscript according to the comments provided. 

Reviewer 4 Report

In this paper authors employ machine learning schemes for the prediction of RSSI in hybrid FSO/RF systems.

All in all the paper addresses a timely topic, however I have the following concerns.

a) A major concern is the quality of presentation. The writing should be further elaborated in several parts of the text and the quality of some figures need to be improved.

b) Please explain your motivations on the use of machine learning for the problem under consideration.

c) Please check carefully the names of the authors in your references.

For example, see Ref. [11]

Reviewer 5 Report

The manuscript 'Investigation of machine learning methods for prediction of 2 measured values of atmospheric channel for Hybrid FSO/RF 3 system' discusses the improvement of decision trees and decision trees using the Ada-Boost regressor algorithms for the precise prediction of the RSSI parameter to enable hard switching in a hybrid FSO/RF system effectively. This manuscript has the potential to be accepted for publication in the Photonics.

Round 2

Reviewer 2 Report

In my view the use of AdaBoost in the estimation of the RSSI do not show a very high performance. The comparision of AdaBoost with other techniques of machine learning such as random forest may increase the appealing of the paper.  The final results are not clearly shown not the scheme that it is used for encoding, and decoding of the data. It is not only the machine learning system that has been developed. It is not explained the signal that you have used, the frames, the equalization procedure.  So, it is better to reject the paper and redo it with this new ideas.

Reviewer 4 Report

Please check authors names in reference 11.